

# UQCRC1 downregulation impairs cognitive function in mice *via* AMPK inactivation

Jing Zhang[1,2], Zuoxi Wu[1], Zonghong Long[1], Feng Ceng[1], Fuhai Bai[1] and Hong Li[1]

[1] Department of Anesthesiology, The Xinqiao Hospital, Army Medical University, Chongqing, China
[2] Department of Anesthesiology, Affiliated Hospital of North Sichuan Medical College, Nanchong, Sichuan, China

## ABSTRACT

**Background.** Ubiquinol-cytochrome c reductase core protein 1 (UQCRC1) is an essential subunit of complex III in the mitochondrial respiratory chain. Although earlier studies have indicated that UQCRC1 downregulation causes cognitive impairment, the underlying mechanisms remain unclear.

**Methods.** To investigate its pathophysiological effects, we developed a mouse model with downregulated UQCRC1 expression. Hippocampus-dependent cognitive performance was evaluated using a series of behavioral paradigms. Mitochondrial bioenergetic status was assessed by measuring adenosine triphosphate (ATP) levels, while oxidative stress was quantified through detection of reactive oxygen species (ROS). Molecular analyses were performed to assess AMP-activated protein kinase (AMPK) signaling dynamics and autophagic flux. Additionally, pharmacological interventions aimed at activating AMPK and enhancing lysosomal function were employed to elucidate mechanistic pathways.

**Results.** Downregulation of UQCRC1 resulted in significant deficits in hippocampus-dependent cognitive performance, accompanied by impaired mitochondrial bioenergetics (lower ATP synthesis) and elevated oxidative stress (increased ROS levels). Mechanistically, these phenotypes were associated with diminished AMPK activation and disrupted autophagic flux. Importantly, pharmacological activation of AMPK or enhancement of lysosomal activity in UQCRC1-deficient mice effectively ameliorated cognitive deficits and restored mitochondrial redox homeostasis .

**Conclusions.** This study identifies AMPK as a pivotal metabolic orchestrator of mitochondrial-lysosomal functional crosstalk and reveals its non-canonical function in maintaining neuronal homeostasis *via* coordinated regulation of autophagic flux and redox balance. Our findings propose AMPK-driven interorganelle communication as a modifiable therapeutic target for addressing cognitive decline resulting from mitochondrial dysfunction.

Corresponding authors
Fuhai Bai, bfh@tmmu.edu.cn
Hong Li, lh78553@tmmu.edu.cn

## INTRODUCTION

Mitochondria are essential organelles in mammalian cells, functioning as central hubs for energy metabolism and signal transduction. They play pivotal roles in a wide array of cellular processes, including proliferation, differentiation, apoptosis, and autophagy (*Zheng et al., 2021*; *Zhou et al., 2018*). A cornerstone of mitochondrial energy production is the electron transport chain (ETC), composed of two electron carriers—ubiquinone and cytochrome c—and four enzyme complexes (Complexes I–IV). Complex III (ubiquinol-cytochrome c reductase), a symmetric dimer with eleven subunits per monomer, includes UQCRC1, a protein essential for its assembly and function (*Fernandez-Vizarra & Zeviani, 2018*). Mitochondrial dysfunction, particularly in neurons, can result in excessive reactive oxygen species (ROS) production, cytochrome c release, diminished adenosine triphosphate (ATP) synthesis, and impaired activity of respiratory complexes. These disruptions ultimately lead to neuronal apoptosis and cognitive deterioration (*Fernandez-Vizarra & Zeviani, 2018*). Mutations in UQCRC1 have been linked to mitochondrial respiratory chain deficiencies and neurodegenerative disorders such as Parkinson's disease, underscoring its critical role in maintaining mitochondrial integrity and neuronal viability (*Lin et al., 2020*). Despite growing evidence linking UQCRC1 to mitochondrial integrity and neurodegeneration, the specific mechanisms by which UQCRC1 influences cognitive function remain poorly understood. In particular, how UQCRC1-mediated mitochondrial dysfunction contributes to ROS accumulation, autophagy disruption, and neuronal apoptosis has not been fully elucidated.

Importantly, mitochondria do not function in isolation but are closely integrated with quality control mechanisms such as autophagy. Autophagy maintains cellular homeostasis by degrading and recycling damaged organelles and proteins (*Mizushima & Komatsu, 2011*). While autophagy generally promotes cell survival, apoptosis is activated when cellular damage exceeds repair capacity. Apoptosis is a genetically regulated, physiological form of cell death that occurs under specific physiological or pathological conditions (*Renehan, Booth & Potten, 2001*; *Bredesen, 1995*). Mitochondria are central to the apoptotic pathway; their dysfunction can initiate apoptosis through multiple routes, including the release of cytochrome c and apoptosis-inducing factor (AIF), as well as excessive opening of the mitochondrial permeability transition pore (mPTP). Lower ATP levels and elevated ROS further sensitize neurons to apoptosis-related proteins, thereby accelerating the apoptotic process (*Nguyen et al., 2023*; *Bock & Tait, 2020*; *Wang et al., 2023*; *Picca et al., 2023*; *Katayama et al., 2020*; *Kerr et al., 2017*).

AMP-activated protein kinase (AMPK) is a key energy sensor and regulator of cellular stress responses. It is activated under conditions such as energy depletion, increased cytosolic Ca2+, and elevated ROS levels (*Trefts & Shaw, 2021*). AMPK not only modulates mitochondrial dynamics by influencing fusion and fission but also promotes mitochondrial biogenesis and supports lysosomal function (*Virga et al., 2024*; *Herzig & Shaw, 2018*; *Paquette et al., 2021*). Given its dual roles in regulating mitochondria and lysosomes, AMPK acts as a crucial link between mitochondrial dysfunction and impaired autophagy (*Hu et al., 2021*).

This study aims to investigate the mechanistic role of UQCRC1 in cognitive regulation. We hypothesize that UQCRC1 deficiency induces mitochondrial dysfunction, characterized by reduced ATP production and increased ROS generation. These changes are expected to impair autophagy, promote neuronal apoptosis, and ultimately lead to cognitive deficits. To test this hypothesis, we employed UQCRC1$^{+/-}$ mice and assessed hippocampus-dependent cognitive function, mitochondrial bioenergetics, ROS levels, autophagy flux, and neuronal apoptosis. Our findings highlight the importance of lysosomal function and AMPK activation in mediating these pathological changes, offering novel insights into the molecular underpinnings of UQCRC1-related cognitive impairment.

## MATERIALS & METHODS

### Animals

C57BL/6 wild-type (WT) mice were obtained from Charles River Laboratories (Chengdu, China), while UQCRC1$^{+/-}$ heterozygous mice were generated by Professor Zhiyi Zuo (*Shan et al., 2019*), with all animals housed under specific pathogen-free (SPF) conditions at Army Medical University's animal facility. Mice aged 8–12 weeks (weighing 20–30 g) were maintained in standard cages ($330 \times 210 \times 170$ mm; 5 mice/cage) with *ad libitum* access to food and water under controlled environmental parameters: 12-h light/dark cycle (08:00–20:00), 20–23 °C ambient temperature, and 50–60% relative humidity.

In addition to 16 female animals (eight UQCRC1$^{+/-}$ and eight WT) designated for behavioral assessments, this study included 36 male WT mice and 63 male UQCRC1$^{+/-}$ mice. Following initial behavioral evaluations, eight WT and eight UQCRC1$^{+/-}$ males were each joined by 10 more genotype-matched males (for a total of $n = 18$ per genotype) for hippocampus collection at baseline. Six specimens per genotype were assigned to ROS/ATP/caspase assays, six to Western blot analysis, and six to transmission electron microscopy (TEM). For the therapeutic evaluation phase, 45 UQCRC1$^{+/-}$ males were randomly assigned into three treatment cohorts (solvent vehicle, A-769662, and LH2-051). Eight mice per group received post-treatment behavioral assessment, while seven additional mice per group were included for tissue analysis (six specimens per group were assigned to ROS/ATP/caspase assays, three to Western blot analysis, and six to TEM). A parallel study included 36 WT males following the same experimental timeline and tissue distribution protocols. Animals were euthanized prior to the planned endpoint only if they met predefined humane criteria, including (but were not limited to): severe weight loss (>20% of baseline body weight) or failure to thrive; signs of irreversible distress or pain (*e.g.*, labored breathing, prolonged immobility, inability to access food/water); unexpected complications directly related to the experimental intervention (*e.g.*, neurological deficits). No animals required early euthanasia in this study, as all subjects maintained stable health metrics within predefined thresholds throughout the experimental timeline. No animals were retained beyond the study period due to the terminal nature of the experimental design. Males were euthanized *via* intraperitoneal injection of 1% sodium pentobarbital (50 mg/kg), while female cohorts were euthanized using gradual $CO_2$ asphyxiation (30%–99%, 15 min) with confirmation of death. Group assignments were randomized using random

number tables. Sample sizes calculated based on power analysis incorporating preliminary data and literature benchmarks (*Shan et al., 2019*; *Lin et al., 2020*; *Fernandez-Mosquera et al., 2019*; *Chen et al., 2022*; *Kim et al., 2021*). The experimental protocol was developed prior to study commencement and conducted in compliance with guidelines approved by the Army Medical University's Laboratory Animal Welfare and Ethics Committee (Approval No.: AMUWEC20245280; Approval date: 10/1/2024).

## Behavioral testing

All mice were acclimated to their environment one week prior to the onset of behavioral trials. All behavioral tests commenced at 10 a.m. To minimize environmental stress, mice were transferred to the testing room two hours before each session. Behavioral assessments were spaced one day apart to prevent interference from prior testing. After each test, all equipment was thoroughly cleaned with 75% ethanol to eliminate olfactory cues. During testing, examiners exited the room to reduce external influence. Behavioral data were recorded and analyzed using EthoVision XT 11.5 (Noldus Inc., Wageningen, Netherlands).

### Novel object recognition test

The ability of short-term memory in the hippocampus of mice was evaluated using the novel object recognition (NOR) test (*Bevins & Besheer, 2006*). The experimental setup consisted of an acrylic rectangular enclosure ($40 \times 40 \times 40$ cm, Fig. 1A). The test comprised three separate phases. Habituation Phase: mice were allowed to explore the empty apparatus freely for 10 min. Familiarization Phase: conducted 24 h after habituation, two identical objects (Familiar Object, F) were placed symmetrically within the apparatus. Mice were reintroduced into the center of the apparatus and allowed to explore for 10 min. Test Phase: two hours after the second stage, one of the familiar items was replaced with a novel object of the same size but a different shape (Novel Object, N). Mice were again placed in the center and given 10 min to explore. The time spent exploring the familiar object (tF) and the novel object (tN) was recorded. The recognition rate (RR) was calculated as RR = tN / (tF + tN).

### Nest building test

The nest building test (NBT) was utilized to assess the impact of UQCRC1 knockdown on hippocampus-dependent behavior. Mice were individually housed in standard cages with a single 2.5 g, 5 $cm^2$ tearable cotton pad and a small amount of wood shavings on the day of the test at 6:00 PM. Food and water were provided *ad libitum*. The following morning, nesting activity was photographed, with particular attention to the degree of shredding and use of the cotton pad (Fig. 1C). Nesting quality was subsequently scored according to criteria defined in earlier research (*Deacon, 2006*). Data from mice whose nesting material was moist were excluded from the analysis.

### Barnes maze

The Barnes maze test was employed as a low-stress and effective way to evaluate spatial learning and memory. The apparatus consisted of a circular platform with 20 evenly spaced holes, only one of which led to an escape box (target box). The escape box provided a

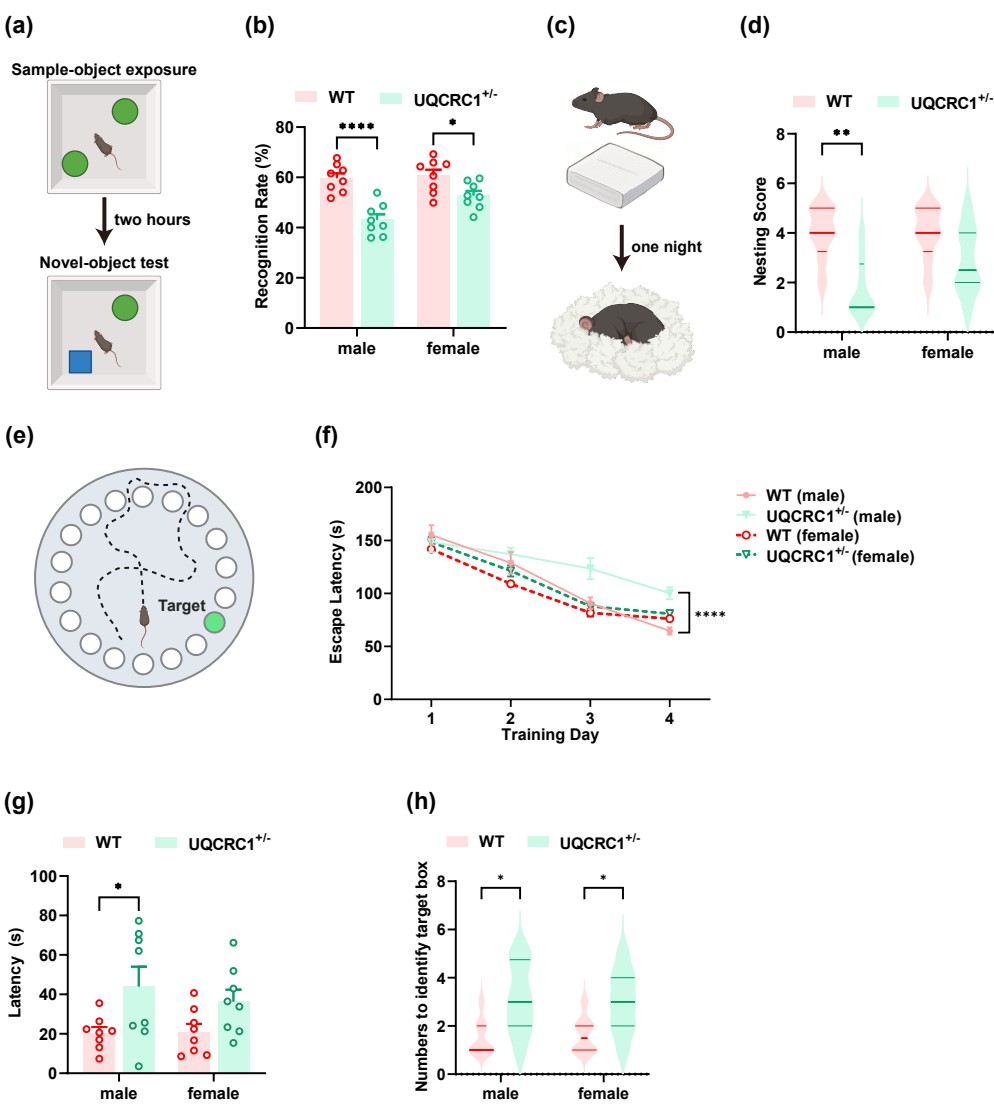

**Figure 1** **The downregulation of UQCRC1 impaired cognitive function.** (A) Schematic representation of NOR, created in https://BioRender.com. (B) The recognition rate of both male and female UQCRC1+/- mice was lower than that of their conspecifics ($n = 8$). (C) Schematic representation of NBT, created in https://BioRender.com. (D) The nesting score of male UQCRC1+/- mice was lower ($n = 8$). (E) Schematic representation of the Barnes maze, created in https://BioRender.com. (F) Compared to male WT mice, male UQCRC1+/- mice exhibited longer escape latencies from day 3 during the training phase of Barnes maze ($n = 8$). (G) Male UQCRC1+/- mice had prolonged escape latencies during the testing phase of the Barnes maze ($n = 8$). (H) Both male and female UQCRC1+/- mice took more attempts to find the correct hole ($n = 8$). Recognition rate in NOR and escape latencies in both the training and testing phases of the Barnes maze were quantified and expressed as mean ±standard error of the mean (SEM). Nesting score in NBT and the numbers to identify target box during Barnes maze were recorded and presented using non-parametric descriptors (median with interquartile range). ****$p < 0.0001$; **$p < 0.01$; *$p < 0.05$.

darkened refuge from aversive stimuli such as bright light and noise (Fig. 1E). The test comprised two phases. Training phase: mice were placed in the center of the platform and

allowed to explore freely. Trials ended when the mouse located and entered the escape box or after three minutes had elapsed. If the mouse failed to locate the box, it was gently guided to the correct location and allowed to stay there for one minute. Each mouse received three training trials per day for four consecutive days. The primary outcome measure was latency to enter the escape box. Testing phase: conducted 24 h after the training session, the testing phase proceeded exactly like the training phase. Performance was evaluated based on the latency to locate the escape box and the number of tries to find the box.

## Transmission electron microscope

Mice were initially perfused with ice-cold phosphate-buffered saline (PBS). The hippocampus was carefully dissected and sectioned into small fragments. The tissue samples were subsequently fixed, dehydrated, infiltrated, and embedded according to standard protocols. Ultrathin sections were prepared and stained with 2% uranyl acetate. Quantitative analysis of the images was performed using ImageJ software (NIH, https://imagej.net/ij/, version 1.54).

## Western blotting

Hippocampal tissues were harvested and homogenized, and 20 μg of protein was loaded onto the 4–20% gradient gels (CAT# ET15420LGel; ACE Biotechnology) and then transferred to a PVDF membrane using the Bio-Rad system. The membranes were subsequently blocked with rapid blocking buffer (CAT# HY-K1027; MedChemExpress) for 10 min. Primary antibodies were incubated overnight at 4 °C. The primary antibodies used included rabbit polyclonal anti-LC3B (1:1000 dilution, Abcam, CAT# ab48394), rabbit polyclonal anti-AMPK$\alpha$ (1:1000 dilution, CAT# 2532; Cell Signaling Technology, Danvers, MA, USA), rabbit monoclonal anti-phospho-AMPK$\alpha$ (Thr172) (1:1000 dilution, CAT# 2535; Cell Signaling Technology, Danvers, MA, USA), and rabbit monoclonal anti-GAPDH (1:3000 dilution, CAT# LF211; Epizyme Biotech, Cambridge, MA, USA). After incubation with HRP-conjugated goat anti-rabbit IgG (1:3000 dilution, CAT# ZB-2301; ZSGB-BIO) for one hour at room temperature, membranes were developed. Densitometric analysis was conducted using ImageJ software (version 1.54). Protein expression levels were normalized by dividing the signal intensity of the target protein by that of GAPDH. All wild-type (WT) group values were set to 1.0 to establish baseline expression.

## Quantitative PCR (qPCR)

Total RNA was extracted from murine hippocampal tissues using the Magbead RNA Extraction Kit (CAT# W3711S; Cwbiotech), followed by genomic DNA removal using RNase-free DNase I (CAT# EN0521; Thermo Fisher Scientific, Waltham, MA, USA). Complementary DNA synthesis and amplification were performed using the PrimeScript RT-PCR Kit (CAT# RR037A; Takara Bio). Quantitative analysis employed the $\Delta\Delta$CT method with normalization to GAPDH expression. Primer sequences are provided in Table S1.

## ATP assay

ATP content was measured following the protocol provided in the ATP content test kit manual (CAT# G4309-48T; Servicebio). Extracted hippocampal tissues were homogenized,

lysed, boiled, and cooled to room temperature, followed by centrifugation at 10,000× g for 15 min at 4 °C. Subsequently, 20 μL of the supernatant was mixed with 100 μL of ATP assay reagent. Bioluminescence intensity was subsequently measured using a luminometer (Fig. 2A).

### Assessment of reactive oxygen species

ROS levels were assessed using the reactive oxygen species (ROS) Detection Kit (Bestbio, CAT# BB-47051). After hippocampal homogenization, the homogenate was centrifuged at 1,000× g for 3 min at 4 °C. A volume of 2 μL dihydroethidium (DHE) probe was added to 200 μL of the supernatant and incubated in the dark at 37 °C for 30 min. Fluorescence intensity was measured using an excitation wavelength of 510 nm and an emission wavelength of 610 nm. ROS levels were determined by calculating the ratio of fluorescence intensity to protein concentration.

### Assessment of caspase 3 and caspase 9

The Caspase 3/Caspase 9 Activity Assay Kit (CAT# APC03/APC09; MultiSciences Biotech Co., Ltd.) was used to evaluate the activation of caspase 3 and caspase 9. Briefly, homogenized hippocampal tissue was centrifuged at 12,000 rpm for 15 min at 4 °C. The supernatant was collected and reaction mixtures were prepared according to the manufacturer's instructions. Following 4 h of incubation at 37 °C, absorbance was measured at 405 nm.

### Intraperitoneal injection

$UQCRC1^{+/-}$ mice were randomly assigned to three groups: $UQCRC1^{+/-}$ + A-769662, $UQCRC1^{+/-}$ + LH2-051, and $UQCRC1^{+/-}$ + solvent. Mice in the $UQCRC1^{+/-}$ + A-769662 and $UQCRC1^{+/-}$ + LH2-051 cohorts received intraperitoneal injections of A-769662 (30 mg/kg, MCE, CAT# HY-50662) or LH2-051 (10 mg/kg, MCE, CAT# HY-161723), administered twice daily for 30 consecutive days (Fig. 3A). Mice in the $UQCRC1^{+/-}$ + solvent group received an equivalent volume of solvent comprising 10% DMSO, 40% PEG300, 5% Tween-80, and 45% saline.

### Statistical analysis

All experiments and analyses were conducted under blinded conditions, with investigators unaware of group allocations during both experimental procedures and data interpretation. All acquired data were retained for statistical analysis without exclusion. The Shapiro–Wilk test was employed to assess data normality. Parametric tests were applied to normally distributed data, while non-parametric tests were applied when normality assumptions were not met. For two-way ANOVA, no outliers were detected based on studentized residuals exceeding ±3. The Kruskal–Wallis $H$ test, unpaired two-tailed $t$-tests, Mann–Whitney $U$ test, one-way ANOVA, and two-way ANOVA were used to assess differences between groups. Statistical significance was defined as a $p$-value of less than 0.05. Data analysis was performed using GraphPad Prism (version 9.5; GraphPad Software, Boston, MA, USA) and SPSS (version 27.0; IBM, Armonk, NY, USA). Statistical significance was indicated as $^*p < 0.05$, $^{**}p < 0.01$, $^{***}p < 0.001$, and $^{****}p < 0.0001$.

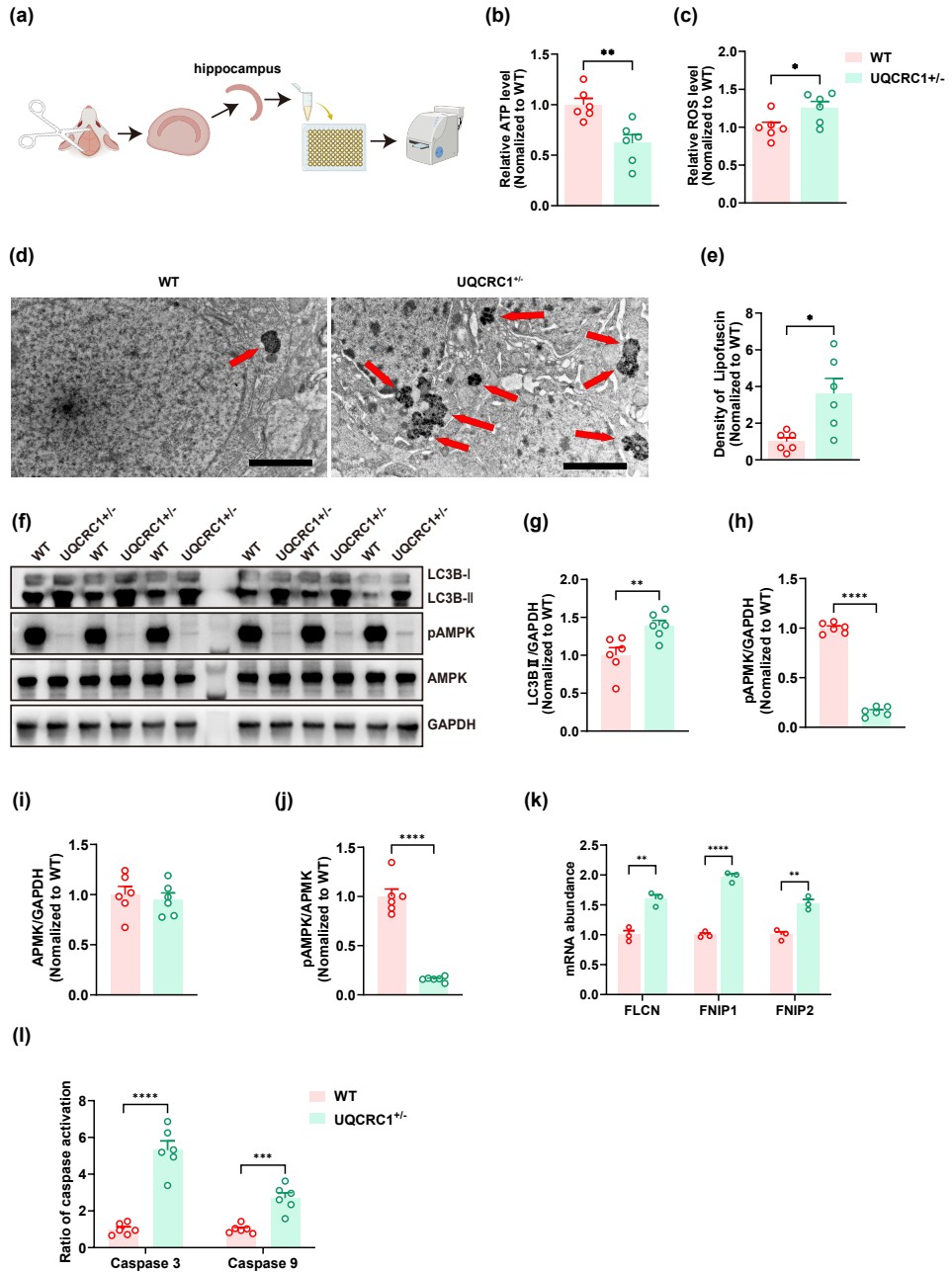

**Figure 2** **The downregulation of UQCRC1 led to autophagy impairment.** (A) Schematic illustration of the experimental design for ATP assays, reactive oxygen species assessments, and caspase 3 and caspase 9 evaluations, created in https://BioRender.com. (B) The ATP level in UQCRC1 +/- mice was lower. (C) The ROS level in UQCRC1 +/- mice was higher. (D) Representative transmission electron microscope (TEM) images of WT mice and UQCRC1 +/- mice (red arrows indicate lipofuscin; scale bar: 2 μm). (E) The density of lipofuscin was higher in UQCRC1 +/- mice. (F) Western blot images of LC3B I, LC3B II, pAMPK, AMPK and GAPDH in WT mice and UQCRC1 +/- mice. (G-J) Quantification of protein expression of LC3B II (G), pAMPK (H), AMPK (I) and pAMPK/AMPK ratio (J). Protein levels were normalized to GAPDH as an internal loading control . (K) Increased mRNA expression of FLCN, FNIP1, and FNIP2 in UQCRC1 +/- mice. (L) Activation of both caspase 3 and caspase 9 was higher in UQCRC1 +/- mice. Data are represented as mean ±SEM. ****$p < 0.0001$; ***$p < 0.001$; **$p < 0.01$; *$p < 0.05$.

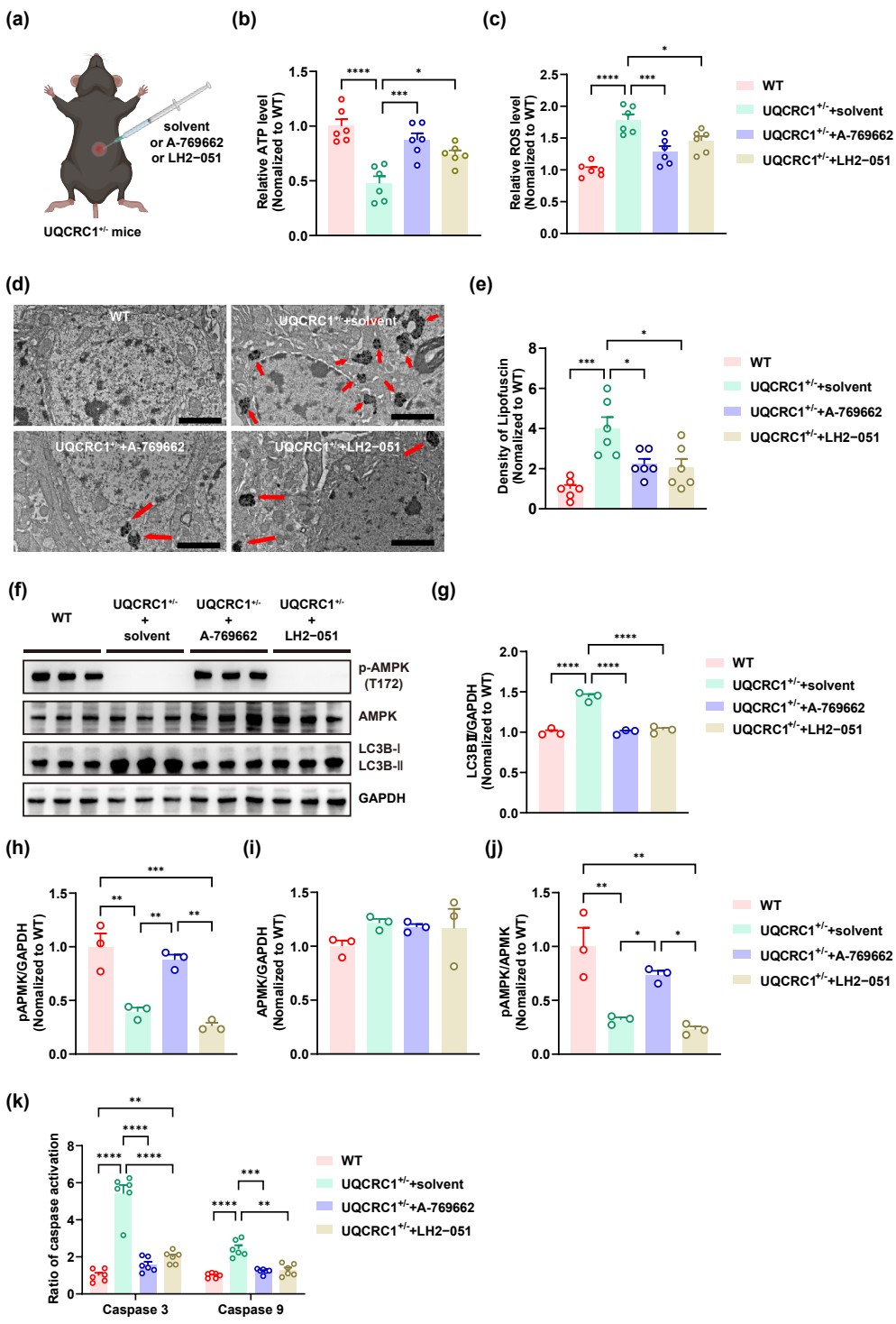

**Figure 3   Both lysosomal function improvement and AMPK activity enhancement ameliorated autophagy in UQCRC1 mice.** (A) Schematic illustration of the experimental design for Intraperitoneal injection, created in https://BioRender.com. (B) The ATP level in UQCRC1+/- mice increased after the (continued on next page...)

**Figure 3 (…continued)**
administration of A-769662 or LH2-051. (C) The ROS levels in UQCRC1+/- mice diminished following the treatment with A-769662 or LH2-051. (D) Representative transmission electron microscope (TEM) images (red arrows indicate lipofuscin; scale bar: 2 μm). (E) The density of lipofuscin in UQCRC1+/- mice reduced after the administration of A-769662 or LH2-051. (F) Western blot image of pAMPK, AMPK, LC3B a and LC3B II, and GAPDH. (G–J) Quantification of protein expression of LC3B II (G), pAMPK (H), AMPK (I), and pAMPK/AMPK (J). Protein levels were normalized to GAPDH as an internal loading control . (K) Following administration of A-769662 or LH2-051, activation of caspase 3 and caspase 9 in UQCRC1+/- mice was reduced. Data are represented as mean ±SEM. **** $p < 0.0001$, *** $p < 0.001$, ** $p < 0.01$, *$p < 0.05$.

## AI tool usage statement

The DeepSeek-R1 model (developed by DeepSeek) was employed exclusively for grammatical error detection, spelling correction, and sentence structure refinement. The tool did not contribute to data generation, conceptual development, or conclusion formulation. All modifications suggested by the tool underwent rigorous manual verification by the authors, who assume full responsibility for the perspectives, data analysis, and scholarly argumentation presented herein.

## RESULTS

### The downregulation of UQCRC1 impairs cognitive function

In the NOR test, both male (Fig. 1B; $p < 0.0001$) and female (Fig. 1B; $p < 0.05$) UQCRC1$^{+/-}$ mice demonstrated a markedly reduced recognition rate compared to WT controls. In the NBT, male UQCRC1$^{+/-}$ mice exhibited impaired performance (Fig. 1D; $p < 0.01$). Starting on day 3 of the Barnes maze training phase, male UQCRC1$^{+/-}$ mice exhibited substantially longer escape latencies (Fig. 1F; $p < 0.05$ on day 3; $p < 0.01$ on day 4). During the testing phase, these mice also demonstrated significantly longer escape latencies (Fig. 1G; $p < 0.05$) and a notably higher number of attempts to find the target box (Fig. 1H; $p < 0.05$). Female UQCRC1$^{+/-}$ mice also required more attempts to locate the target box (Fig. 1H; $p < 0.05$).

These results suggested that mitochondrial dysfunction caused by UQCRC1 downregulation is strongly correlated with cognitive deficits in male mice, whereas its impact on female mice appeared to be more variable. Consequently, subsequent investigations focused primarily on male mice.

### The downregulation of UQCRC1 leads to autophagy impairment

Our findings showed that the downregulation of UQCRC1 significantly reduced hippocampus-dependent cognitive functions. To further investigate this phenomenon, we assessed cellular energy status and oxidative stress levels. A significant decrease in ATP levels was observed in UQCRC1$^{+/-}$ mice (Fig. 2B; $p < 0.01$), accompanied by an increase in ROS levels (Fig. 2C; $p < 0.05$). Lipofuscin, a yellowish-brown granular pigment that accumulates in ageing cells, is formed through the cross-linking of incompletely degraded lipids and proteins. Its accumulation serves as an indicator of impaired lysosomal degradation capacity. In our study, increased lipofuscin accumulation was detected (Figs. 2D, 2E; $p < 0.05$), along with elevated expression of LC3B-II (Figs. 2F, 2G; $p < 0.01$). These results

implied that UQCRC1 downregulation results in defective autophagic flux. Consistent with previous reports (*Fernandez-Mosquera et al., 2019*), our data showed a significant reduction in pAMPK levels (Figs. 2F, 2H; $p < 0.0001$) and in pAMPK/AMPK ratio (Figs. 2F, 2J; $p < 0.0001$), whereas total AMPK levels remained unchanged (Figs. 2F, 2I). To date, folliculin (FLCN), a known tumor suppressor, is the only identified direct negative regulator of AMPK activity. FLCN forms a functional heterotrimeric complex with its interacting proteins FNIP1 and FNIP2, which together inhibit AMPK signaling (*Xiao et al., 2021*). Notably, previous studies have demonstrated upregulation of FLCN, FNIP1, and FNIP2 in UQCRC1 knockdown cells (*Fernandez-Mosquera et al., 2019*). In alignment with these findings, our study confirmed the transcriptional upregulation of FLCN (Fig. 2K; $p < 0.05$), FNIP1 (Fig. 2K; $p < 0.05$), and FNIP2 (Fig. 2K; $p < 0.05$). Together, these findings indicate that UQCRC1 downregulation reduced AMPK activation in the hippocampus and impaired autolysosomal degradation. Moreover, we observed increased activation of caspase 3 (Fig. 2I; $p < 0.0001$) and caspase 9 (Fig. 2I; $p < 0.001$), suggesting enhanced apoptotic activity as a consequence of UQCRC1 downregulation. Notably, no differences were observed in ATP levels, ROS levels, or the expression of pAMPK, AMPK, and LC3B between female UQCRC1+/− mice and female WT mice (Fig. S1).

## Enhancing AMPK activity or improving lysosomal function ameliorates autophagy in UQCRC1$^{+/-}$ mice

A-769662 is a direct, allosteric activator of AMPK that increases its activity through two complementary mechanisms: by inhibiting Thr-172 dephosphorylation and directly inducing allosteric activation of AMPK (*Cool et al., 2006*; *Göransson et al., 2007*). LH2−051 improves lysosomal function through the DAT-CDK9-TFEB signaling pathway, thereby promoting the degradation of toxic protein aggregates such as $\beta$-amyloid (*Yin et al., 2023a*; *Yin et al., 2023b*). In this study, we investigated the effects of these compounds in UQCRC1$^{+/-}$ mice. Following intraperitoneal administration of A-769662, we observed a significant restoration in ATP content (Fig. 3B; $p < 0.001$), a reduction in ROS levels (Fig. 3C; $p < 0.001$), and decreased lipofuscin accumulation (Figs. 3D, 3E; $p < 0.05$). Furthermore, AMPK activity (Figs. 3F, 3H–3J; $p < 0.01$) was notably increased, alongside a marked reduction in the LC3B-II expression (Figs. 3F, 3G; $p < 0.0001$), activation of caspase 3 (Fig. 3K; $p < 0.0001$) and caspase 9 (Fig. 3K; $p < 0.001$). In parallel, LH2-051 administration led to significant improvements in ATP content (Fig. 3B; $p < 0.05$), reductions in ROS levels (Fig. 3C; $p < 0.05$), and decreased lipofuscin deposition (Figs. 3D, 3E; $p < 0.05$). LC3B-II expression (Figs. 3F, 3G; $p < 0.0001$) and caspase 9 activation (Fig. 3K; $p < 0.01$) were also restored. Although caspase 3 levels showed partial recovery (Fig. 3K; $p < 0.0001$), they did not return to the levels observed in the control group. Notably, no significant changes were detected in AMPK expression, pAMPK expression, or the pAMPK/AMPK ratio (Figs. 3F, 3H–3J).

Collectively, these findings confirm that lysosomal dysfunction is closely associated with mitochondrial impairment in the hippocampal tissue of UQCRC1-deficient mice. Both AMPK activation (*via* A-769662) and lysosomal enhancement (*via* LH2-051) alleviate these deficits, highlighting their therapeutic potential. However, the incomplete functional

rescue suggests the involvement of additional pathological mechanisms beyond AMPK and lysosomal pathways.

### Increasing AMPK activity or enhancing lysosomal function can both rescue cognitive deficits in UQCRC1$^{+/-}$ mice

To investigate the potential causal relationship between autophagy dysfunction and the cognitive abnormalities observed in UQCRC1$^{+/-}$ mice, we performed a series of behavioral evaluations following rescue therapies. The results demonstrated that augmenting AMPK activity significantly improved the performance of UQCRC1$^{+/-}$ mice in the novel object recognition test (Figs. 4A, 4B; $p < 0.001$), the nest-building test (Figs. 4C, 4D; $p < 0.01$), and the Barnes maze (Figs. 4E–4H).

Enhancing lysosomal function partially ameliorated the cognitive deficits in UQCRC1$^{+/-}$ mice. Specifically, treated mice exhibited a higher recognition rate in the novel object recognition test (Figs. 4A, 4B; $p < 0.01$), as well as reduced latency during both the training phase (Fig. 4F; $p < 0.01$) and testing phase (Fig. 4G; $p < 0.05$) of the Barnes maze. However, no significant improvement was observed in the nesting score during the nest-building test (Figs. 4C, 4D), nor in the number of attempts required to locate the target box in the Barnes maze (Fig. 4H).

These findings indicate that AMPK plays a pivotal role in the cognitive impairments associated with mitochondrial dysfunction, and that lysosomal dysfunction is also implicated in this pathological process.

## DISCUSSION

Because UQCRC1$^{-/-}$ mice displayed embryonic lethality (*Shan et al., 2019*), heterozygous UQCRC1+/− mice were therefore chosen as the experimental model. In this study, our results demonstrate that downregulation of UQCRC1 expression led to decreased ATP levels and increased oxidative stress in hippocampal tissue. Furthermore, this reduction impaired autophagic flux by attenuating AMPK activity, which in turn increased neuronal apoptosis and contributed to hippocampus-dependent cognitive dysfunction.

Sex differences in vulnerability to cognitive impairment due to mitochondrial dysfunction have been well documented (*Bigio et al., 2025*; *Silaidos et al., 2018*). Pre-menopausal females often exhibit stronger antioxidant defenses, attributed to estrogen's protective effects (*Viña & Borrás, 2010*; *Mandal, Tripathi & Sugunan, 2012*; *Grimm, Mensah-Nyagan & Eckert, 2016*). Consistent with these findings, we observed that UQCRC1 downregulation had a smaller and less consistent impact in female mice. Therefore, to minimize confounding variables, only male mice were included in the mechanistic investigations.

Mitochondrial dysfunction has been increasingly recognized as a key contributor to cognitive decline in various human neurodegenerative disorders (*Wen et al., 2025*; *Kathiresan et al., 2025*; *Wang et al., 2020*; *Bishop, Lu & Yankner, 2010*). Given the brain's high oxygen dependency and energetic demands, robust mitochondrial function is essential, particularly for hippocampal neuron integrity and cognitive performance (*Watts, Pocock & Claudianos, 2018*; *Khacho, Harris & Slack, 2019*; *He et al., 2022*). Our current

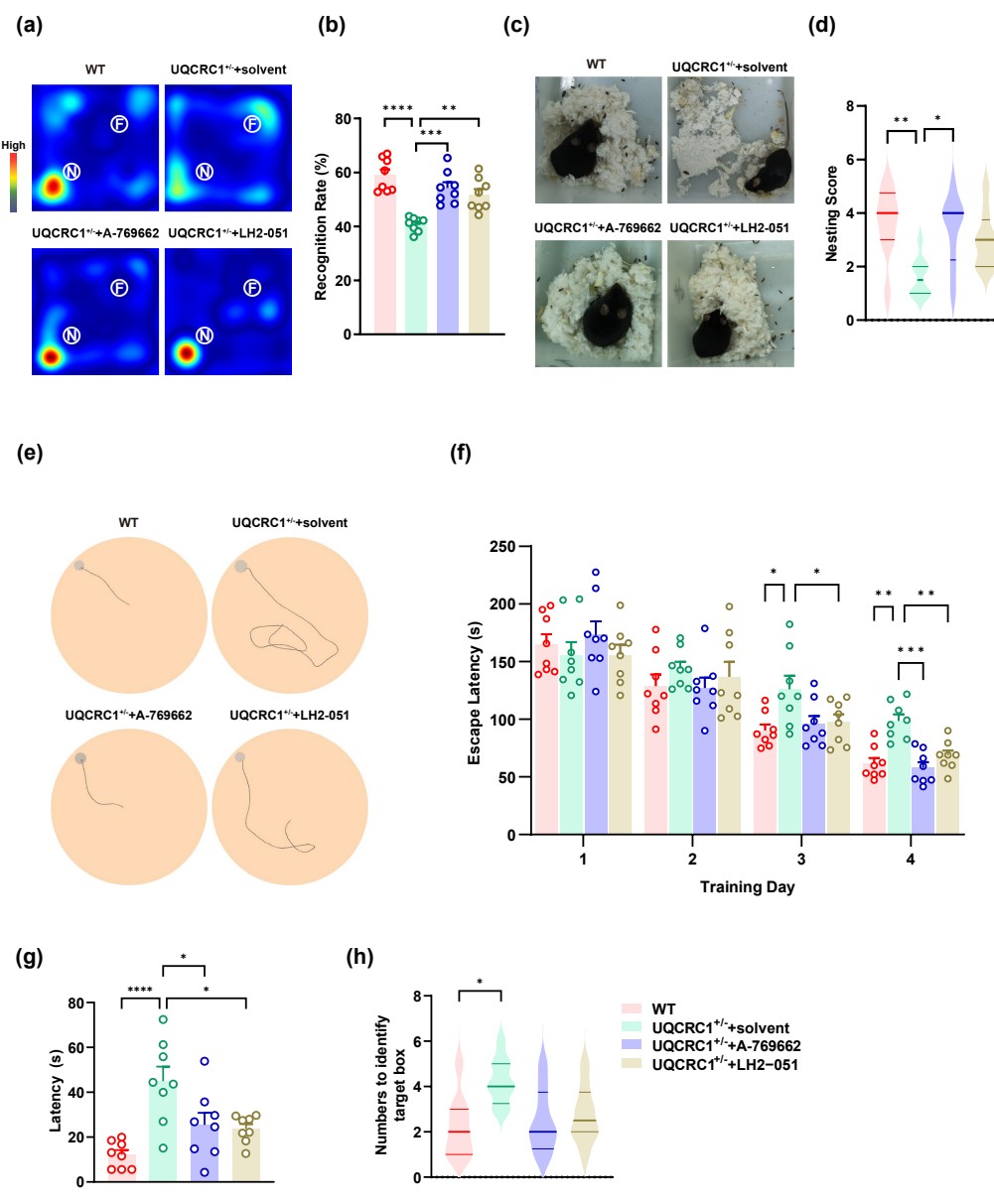

**Figure 4** **The cognitive deficits in UQCRC1+/- mice were ameliorated by augmenting AMPK activation and improving lysosomal function.** (A) Representative heat maps showing the duration and location of the subject during novel object recognition ("N" and "F" represent novel object and familiar object, respectively). (B) The recognition rate of UQCRC1+/- mice increased after the administration of A-769662 or LH2-051. (C) Representative figures of the nest-building test. (D) The nesting score of UQCRC1+/- mice increased after the administration of A-769662. (E) Representative trajectory chart during testing phase of the Barnes maze (gray circle indicates the target hole). (F) Over the training phase of the Barnes maze, the escape latencies of UQCRC1+/- mice were restored to levels similar to WT mice. (G) A-769662 or LH2-051 treatment lowered the escape latencies of UQCRC1+/- mice during the Barnes maze testing phase. (H) UQCRC1+/- mice treated with A-769662 or LH2-051 showed fewer attempts to find the target hole. Recognition rate in NOR and escape latencies in both the training and testing phases of the Barnes maze were quantified and expressed as mean ±SEM. Nesting score in NBT and the numbers to identify target box during Barnes maze were presented as median and interquartile range. **** $p < 0.0001$, *** $p < 0.01$, ** $p < 0.01$, *$p < 0.05$.

study highlights these concepts by demonstrating that the downregulation of UQCRC1, a vital subunit of mitochondrial respiratory chain complex III, significantly impairs hippocampal-dependent cognitive abilities in mice. Moreover, post-mortem analyses of Alzheimer's disease (AD) brains consistently reveal mitochondrial abnormalities and ROS overproduction (*Mutisya, Bowling & Beal, 1994*); these impairments mirror our model's bioenergetic deficit.

Our model specifically revealed mechanistic insights into the cognitive decline, showing a critical role for reduced AMP-activated protein kinase (AMPK) activation and impaired autophagic flux—both hallmark features observed in human Alzheimer's disease (AD) pathology (*Cai et al., 2012*; *Orr & Oddo, 2013*). AMPK is a highly conserved kinase and a critical responder to mitochondrial stress, sensitive to shifts in the AMP:ATP ratio. *Fernandez-Mosquera et al. (2019)* reported that UQCRC1 knockdown reduced AMPK activity in HeLa cells, and similar findings were observed with ndufs4 (a subunit of respiratory chain complex I) knockdown in brain tissue. Furthermore, prior research has demonstrated that AMPK plays a pivotal role in lysosome formation and function *in vitro* and *in vivo* (*Cheng et al., 2021*; *Alers et al., 2012*; *Patra, Weerasekara & Bardeesy, 2019*). In the AD brain, diminished AMPK activity is associated with increased amyloid-beta ($A\beta$) deposition and tau hyperphosphorylation (*Cai et al., 2012*; *Mary et al., 2025*). Simultaneously, compromised lysosomal degradation contributes to the accumulation of damaged mitochondria. Collectively, these findings imply that dysregulation of the AMPK-autophagy axis, driven by UQCRC1 deficiency, could be a convergent pathway underlying cognitive impairments in multiple etiologies. Therefore, our study not only elucidates the AMPK-autophagy axis as a convergent mechanism for cognitive impairment but also strengthens the rationale for developing therapies focused on restoring interorganelle communication to address mitochondrial dysfunction-related cognitive disorders.

A-769662 ($C_{20}H_{12}N_2O_3S$; molecular weight: 360.39 g/mol) is a synthetic compound known for its potent and reversible AMPK activation. Due to its specificity and efficacy, A-769662 is widely used in research exploring cellular energy regulation and potential AMPK-targeted therapies. In animal models, typical dosages range from 10 to 60 mg/kg, with 30 mg/kg commonly used (*Cool et al., 2006*; *Ma et al., 2017*; *Guma et al., 2015*; *Rameshrad et al., 2016*). In our study, although AMPK activity in UQCRC1$^{+/-}$ mice remained lower than in wild-type (WT) controls following A-769662 treatment, the difference was not statistically significant. Concurrently, lysosomal dysfunction and cognitive impairment were alleviated, supporting the potential of A-769662 as a treatment for mitochondrial dysfunction-induced cognitive disorders.

LH2-051 ($C_{27}H_{34}N_2O_3$; molecular weight: 434.57 g/mol) is a small-molecule dopamine transporter (DAT) inhibitor that promotes DAT translocation from the cell membrane to lysosomal membranes. This process activates the protein kinase C (PKC) signaling pathway and modulates transcription factors TFEB and ZKSCAN3, thereby promoting lysosomal biogenesis. A 10 mg/kg dose of LH2-051 significantly improved learning, memory, and cognitive function in AD mouse models by enhancing lysosomal degradation (*Yin et al., 2023a*; *Yin et al., 2023b*). In our study, LH2-051 significantly mitigated cognitive deficits in UQCRC1$^{+/-}$ mice, while AMPK activity showed no significant changes, suggesting

that lysosomal dysfunction is a key factor in mitochondrial damage-induced cognitive decline. Notably, LH2-051 increased ATP levels by 57%, decreased ROS levels by 23%, and reduced Caspase 3 levels by 65% in UQCRC1$^{+/-}$ mice. These results indicate that lysosomes are vital for maintaining cellular homeostasis through ROS regulation and ATP production. However, due to persistent upstream mitochondrial damage, full functional restoration was not achieved. Even after lysosomal function improved in the hippocampus of UQCRC1$^{+/-}$ mice, additional mechanisms may still drive cellular dysfunction.

Autophagy plays a nuanced and essential role in neurons. While it helps maintain neuronal balance and function, excessive or dysregulated autophagy may be detrimental (*Nixon & Rubinsztein, 2024*). LC3II, a structural protein formed during autophagy initiation, associates with autophagosome membranes and is ultimately degraded in autolysosomes (*Iriondo et al., 2022*). Its expression correlates with autophagosome/autolysosome abundance. In this study, we observed increased LC3II expression in the hippocampal tissue of UQCRC1$^{+/-}$ mice. Consistent with this finding, transmission electron microscopy revealed a significant accumulation of lipofuscin in the same region. These observations indicate that autolysosomal degradation and recycling are impaired in the hippocampus of UQCRC1$^{+/-}$ mice, disrupting the autophagic process.

## Limitation

This study has several limitations that warrant consideration. First, our findings suggest sex-dependent differences in the effects of UQCRC1 downregulation on murine cognitive function, yet the underlying mechanisms remain unclear. Second, although behavioral assessments preliminarily identified the hippocampus as the primary region affected, the extensive neuronal networks involved in cognition prevent the complete exclusion of contributions from extra-hippocampal areas. Third, while we demonstrated that UQCRC1-mediated mitochondrial dysfunction impairs cognition through AMPK-dependent autophagic dysregulation, two critical mechanistic gaps persist: (1) the precise signaling cascades linking AMPK activation to autophagic modulation, and (2) the exact neurobiological mechanisms through which autophagy perturbations mediate cognitive deficits.

## CONCLUSIONS

This study demonstrates that UQCRC1 deficiency in mouse models leads to hippocampus-dependent cognitive impairment. Subsequent analyses revealed that UQCRC1 downregulation triggers a series of pathological changes in hippocampal neurons, including reduced ATP production, decreased AMPK activation, increased ROS and lipofuscin accumulation, and enhanced activation of caspase-3 and caspase-9. Through pharmacological intervention in UQCRC1$^{+/-}$ mice, we observed that lysosomal enhancement *via* LH2-051 and AMPK activation through A-769662 effectively mitigated cognitive deficits. Notably, AMPK activation significantly reduced lipofuscin accumulation, whereas lysosomal enhancement had minimal impact on AMPK activity. These findings outline a molecular pathway wherein UQCRC1 deficiency-induced mitochondrial dysfunction exacerbates neuronal apoptosis *via* lysosomal impairment, with AMPK

functioning as a central regulatory hub. This cascade highlights a novel therapeutic target for alleviating cognitive impairment associated with mitochondrial disorders.

## ACKNOWLEDGEMENTS

We are grateful to Professor Zhiyi Zuo's laboratory for providing the UQCRC1+/- mice. The DeepSeek-R1 model was used exclusively for grammar, spelling, and sentence-structure refinement during the manuscripts preparation. The tool played no role in data generation, conceptualization, analysis, or interpretation of results. All AI-suggested edits were rigorously reviewed and verified by the authors, who retain full responsibility for the scholarly content, perspectives, and conclusions presented.

### Funding

This study was supported by the General Program of National Natural Science Foundation of China (No. 82171265), North Sichuan Medical College University-Level Research Development Project (CBY20-QA-Y07), National Natural Science Foundation of China (No. 82101273), Second Affiliated Hospital of Army Medical University Incubation Program for Young Doctoral Talents (No. 2023YQB007). The funders had no role in study design, data collection and analysis, decision to publish, or preparation of the manuscript.

### Grant Disclosures

The following grant information was disclosed by the authors:
General Program of National Natural Science Foundation of China: 82171265.
North Sichuan Medical College University-Level Research Development Project: CBY20-QA-Y07.
National Natural Science Foundation of China: 82101273.
Second Affiliated Hospital of Army Medical University Incubation Program for Young Doctoral Talents: 2023YQB007.

### Competing Interests

The authors declare there are no competing interests.

### Author Contributions

- Jing Zhang conceived and designed the experiments, performed the experiments, analyzed the data, prepared figures and/or tables, and approved the final draft.
- Zuoxi Wu performed the experiments, analyzed the data, prepared figures and/or tables, and approved the final draft.
- Zonghong Long performed the experiments, analyzed the data, prepared figures and/or tables, and approved the final draft.
- Feng Ceng performed the experiments, analyzed the data, prepared figures and/or tables, and approved the final draft.
- Fuhai Bai conceived and designed the experiments, analyzed the data, authored or reviewed drafts of the article, and approved the final draft.

- Hong Li conceived and designed the experiments, authored or reviewed drafts of the article, and approved the final draft.

## Animal Ethics

The following information was supplied relating to ethical approvals (i.e., approving body and any reference numbers):

The research was carried out in line with guidelines authorized by the Army Medical University's Laboratory Animal Welfare and Ethics Committee (AMUWEC20245280).

## Data Availability

Raw data is available in the Supplemental Files.

## Supplemental Information

Supplemental information for this article can be found online at http://dx.doi.org/10.7717/peerj.19873#supplemental-information.

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
