# Peer review of "UQCRC1 downregulation impairs cognitive function in mice *via* AMPK inactivation"

_PeerJ, doi:10.7717/peerj.19873_

## Round 0.1 · original submission · Major Revisions

The manuscript presents data describing a mechanism through which UQCRC1 down-regulation impacts on cognitive dysfunction. The data are convincing and potentially novel, but the reviewers feel that in order for it to become publishable, it needs some revisions.

**Language Note:** The review process has identified that the English language must be improved. PeerJ can provide language editing services - please contact us at [email protected] for pricing (be sure to provide your manuscript number and title). Alternatively, you should make your own arrangements to improve the language quality and provide details in your response letter. – PeerJ Staff

·

Basic reporting

Minor revision required. Changes and suggestions PDF file.

Experimental design

-

Validity of the findings

-

Reviewer 2 ·

Basic reporting

The manuscript is written clearly and mostly without grammar errors. The authors referenced previous research with sufficient details.

Experimental design

The experimental design is clearly drafted in Materials and Methods. Sufficient details were provided on how experiments were set up, and the use of Biorender diagrams helped explain the experiment set-up as well. Research questions were well-defined and meaningful. The topic filled the knowledge gap in UQCRC1's function in cognitive function.

Validity of the findings

Although the data (figures and tables) provided were robust, certain marks used in Figure 3 were not annotated. Besides, the whole of Figure 3 was left undiscussed, where the authors repeated themselves in Figure 2.

Additional comments

Major Issues:
1. The significance of the study is not clear from the introduction. What’s the knowledge gap in studying UQCRC1? It can be improved by focusing on cognitive function, mitochondrial function in ATP generation, ROS buildup, and the link between autophagy and mitochondria.
2. The logic link between the first two paragraphs in the Introduction is not clear. Why is autophagy introduced? The answer is not clear until reading the third paragraph. Consider reorganizing the introduction contents.
3. Lines 267-281 are not a description for Figure 3 but repeat the previous paragraph.
4. If gender disparity is significant in cognitive disorders, why did only Figure 1 have the experimental setup with male vs. female? Do female mice also show any upregulated/downregulated protein expression? Including them as supplemental figures could enhance the significance of the gender disparity argument.
5. LH2-051-treated mice still showed a significant increase in ROS and Caspase 3 expression level. What is the hypothesis of the LH2-051 failure in regulating these?
6. LH2-051 does not show increased pAMPK level, including the discussion is important for readers to understand the hypothesized functioning pathway.
7. A-769662 was also shown to reduce autophagy and cognitive deficiency. But it was not discussed in the discussion.
8. A-769662 does not fully recover pAMPK level. Do you think UQCRC1 regulates AMPK activity through some other pathways?
9. Lines 329-340 can be moved to the Introduction
10. How does UQCRC1 affect the AMPK pathway? Does it physically interact with any of the proteins being tested in the manuscript?

Minor Issues:
1. Line 87 missing reference.
2. Fig. 1A was not cited in the manuscript
3. Line 200: Fig 2A cited twice
4. Line 258, the discussion about the previous study in human cells should be moved into a discussion
5. Indexing of each paragraph should be consistent
6. What’s the “#” in Fig. 3B, E, G, H, I, and J? Explain the use of the mark in statistical analysis or consider removing them.
7. Figure 3 panel should be aligned with Figure 2 panel for easy comparison

Reviewer 3 ·

Basic reporting

The paper is well structured and provides sufficient background and literature. Nevertheless, there are typos, and the use of the English language needs to be enhanced, as verb tenses are not consistent (e.g., in abstract line 34, "our results show" should be "our results showed" to be consistent with past tense).

The term "loss" to describe the reduction of UQCRC1 in heterozygous mice is confusing. It could be interpreted as a complete knockout. Reduction or downregulation would be more accurate and consistent with the nomenclature of UQCRC1 +/− mice.

Minor composition, typographical, or formatting errors crop up here and there that would require fixing in a thorough proofread.

Experimental design

The study design is comprehensive and technically sound, with appropriate animal models and analysis.

But data regarding the two drugs, LH2-051 and A-769662, are lacking: their mechanism of action, reason for their administration, and dosage rationale must be explained more in the Methods section.

In Discussion (line 323), the authors state that LH2-051 increases lysosomal function, but do not indicate what LH2-051 is or how it works. Explaining briefly its mechanisms would enable readers to follow the reasoning and therapeutic echoes. Also missing discussion regarding findings with A-769662.

In the Discussion (line 292), the paper states: "Since UQCRC1-/- mice displayed embryonic lethality." Whether this observation is reported by the authors or inferred from previous literature (quite likely Shan et al., 2019) is unknown. If the latter, a citation must be added here to properly attribute the observation.

Figures are usually easy to read, but figure legends and Western blot analysis require explanation. In particular, for Figures 2F–K, the normalization strategy is not described in the Methods (e.g., was GAPDH used as a housekeeping control?). Information about the antibodies (housekeeping control, dilutions, and methods for quantifying the blot) should be documented in detail to enable reproducibility. Although this can be inferred from raw data, readers shouldn't have to infer this from supplements; a clearer description in the Methods and/or figure legends should be provided.

Validity of the findings

Consider the inclusion of further translational context in the Discussion by relating mitochondrial dysfunction to established human cognitive diseases.

---

## Round 0.2 · Minor Revisions

As you will see, the reviewers agree that the manuscript is now considerably improved. They have noted a few minor issues that need to be rectified prior to acceptance.

Reviewer 2 ·

Basic reporting

The revision addressed previous-raised questions well. I believe the manuscript can benefit the field of research and a broader readership as well.

Experimental design

The experimental design has no flaw.

Validity of the findings

The findings are described clearly in the manuscript.

Additional comments

Some other issues/questions need to be addressed to further strengthen the manuscript:
1. 2nd paragraph of discussion contains a logic flaw: the female mice showed lower significant impairment in cognitive function which could be hypothesized/reasoned to line 1539-1540. But the next sentence is about male mice?
2. The 3rd and 4th paragraphs of discussion talk extensively about known findings in human, but rarely link to the findings in the current research
3. 5th and 6th paragraphs of discussion introduced two drugs used well, however, this should be introduced to some extent in results as well
4. Any consideration about the potential synergistic effect of LH2-051 and A-769662? How would the use of two drugs simultaneously affect AMPK activity and lysosomal degradation, and cognitive function? Any interaction between AMPK activity pathway and lysosomal degradation pathway?
5. Line 533 and Line 553, Line 1006: Section needs reformatting
6. Line 1309, A-769662 and LH2-051 need better introduction in their working mechanisms and the sentence is missing reference

Annotated reviews are not available for download in order to protect the identity of reviewers who chose to remain anonymous.

Reviewer 3 ·

Basic reporting

The manuscript is well written and clearly structured, with appropriate literature background and sufficient context. The authors have addressed prior concerns regarding clarity of language and typographical consistency. They have made edits to ensure consistent verb tense usage and improved formatting. Figure presentation is adequate, and raw data are shared.

Experimental design

The research question is relevant and well defined. The authors have employed a technically sound design, using appropriate models and assays to investigate mitochondrial dysfunction and cognitive impairment in UQCRC1+/- mice. My prior concerns regarding the rationale and description of the two pharmacological agents (LH2-051 and A-769662) have been fully addressed. Mechanisms of action and dosing rationales are now clearly stated in the revised Methods and Discussion sections. The additional references and explanation provide needed translational context.

The Western blot methods and figure legends have been updated to specify normalization strategies (e.g., GAPDH as loading control), antibody details, and densitometric analysis, which improves methodological reproducibility.

Validity of the findings

The conclusions are supported by the data presented. The authors provide sufficient raw data, statistical comparisons, and validation of their findings. I appreciate their response clarifying the origin of the statement on UQCRC1-/- lethality, which is now properly cited. The revised discussion also includes additional translational context, linking mitochondrial dysfunction to human cognitive diseases.

Overall, the findings are scientifically sound and appropriately interpreted within the scope of the results.

Additional comments

No additional comments.

---

## Round 0.3 · accepted · Accept

The authors have now addressed all comments, and the paper is ready for publication.

Reviewer 2 ·

Basic reporting

The revision addressed all previous-raised questions and issues accordingly.

Experimental design

No comment

Validity of the findings

No comment

Additional comments

The manuscript now equips with sufficient findings and appropriate explanation/hypotheses that makes it good to be published.